# Cognitive Functions in Scuba, Technical and Saturation Diving

**DOI:** 10.3390/biology12020229

**Published:** 2023-01-31

**Authors:** Rita I. Sharma, Anna B. Marcinkowska, Natalia D. Mankowska, Monika Waśkow, Jacek Kot, Pawel J. Winklewski

**Affiliations:** 1Department of Human Physiology, Medical University of Gdansk, 80-210 Gdansk, Poland; 2Applied Cognitive Neuroscience Lab, Department of Human Physiology, Medical University of Gdansk, 80-210 Gdansk, Poland; 32-nd Department of Radiology, Medical University of Gdansk, 80-210 Gdansk, Poland; 4Institute of Health Sciences, Pomeranian University in Slupsk, 76-200 Slupsk, Poland; 5National Centre for Hyperbaric Medicine, Institute of Maritime and Tropical Medicine in Gdynia, Medical University of Gdansk, 81-519 Gdynia, Poland

**Keywords:** diving, cognition, high-pressure neurological syndrome

## Abstract

**Simple Summary:**

Self-contained underwater breathing apparatus (scuba) diving is a popular water activity. In our review, we collect data concerning cognitive function impairment in recreational, technical and saturation diving. Cognitive functions such as alertness, memory and decision making seem to be crucial for divers’ safety. More research should be conducted to clearly define the degrees of diving influence on cognitive functions. We present the collected data following division into acute and chronic effects of diving on cognitive functioning. In addition, we describe high-pressure neurological syndrome (HPNS) as deep dives are becoming commonplace. Our summary gathers the available data concerning impaired cognitive functioning in diving. There is a need for more research to be performed to allow further conclusions to be made in this field.

**Abstract:**

Scuba diving as a recreational activity is becoming increasingly popular. However, the safety of this activity, especially in the out-of-comfort zone, has been discussed worldwide. The latest publications bring conclusions regarding negative effects on cognitive functions. We compare the acute and chronic effects of diving on cognitive functioning depending on the type of dive performed, including recreational, technical and saturation diving. However, the results of research show that acute and chronic effects on cognitive functions can be negative. While acute effects are reversible after the ascent, chronic effects include white matter lesions in magnetic resonance imaging scans. We believe that more investigations should be performed to determine the chronic effects that could be observed after a few months of observations in a group of regular, intense divers. In addition, publications referring to technical divers are very limited, which is disquieting, as this particular group of divers seems to be neglected in research concerning the effects of diving on cognitive functions.

## 1. Introduction

Scuba diving, as a common water activity, is associated with diving accidents. According to Buzzacott et al., however, the number of underwater fatalities has appeared to decrease recently. While data are still being collected and analyzed, the exact number of divers worldwide is difficult to estimate [1].

Despite diving being a physical activity, underwater work should also not be neglected as saturation divers spend a couple of hours up to a few days working in caissons. The safety of this activity has therefore been considered due to the fact that diving accidents are still occurring worldwide.

The performance of scuba diving was first reported in the 19th century, as were the possible hazardous effects of diving. Since the 1970s, the number of reported reviews has been increasing [2,3,4,5].

One of the most well-known side effects of diving is decompression sickness (DCS). The first symptoms of this are headache, dizziness, itching and junction pain due to nitrogen accumulation in the form of bubbles during the changed conditions of pressure. Strauss et al. [6] and Boettger [5] published reports describing the noxious effects which were mostly focused on decompression sickness, also called divers’ disease. The etiology of this illness comes with physics and human physiology. DCS can affect all organs, but neurological DCS involving the brain, the spinal cord or both is the most severe form. Because changes in neurological function due to gas bubbles can significantly affect neurological function, symptomatic DCS due to overload with gas bubbles has intentionally not been included in the scope of this publication; this study only focuses on cognitive processes.

The mental processes involved in receiving, storing and using information to direct behavior are referred to as cognitive processes. In essence, they are the capacity to perceive, react, process, comprehend, store and retrieve information, as well as to decide and respond appropriately. The term “cognition” refers to the entire collection of mental processes and abilities that include knowledge, understanding, working memory, attention, judgment and evaluation. It also includes reasoning and “computation”, problem solving and decision making. These abilities are crucial in diving safety. Neuropsychologists can assess cognitive functions with tests such as the Stroop Task, Ruff Figural Fluency Test, California Verbal Learning Test, Rey Auditory Verbal Learning Test, Visual Memory Test and Critical Flicker-Fusion Frequency Test [7,8].

Obviously, cognitive functions such as alertness, sensing, time of reaction, perception, memory, learning, thinking and decision making are crucial for diving safety and successful ascent. Nevertheless, environmental factors as well as diver-dependent factors may influence performance. Cold water and water visibility, depending on the water reservoir, may affect the diver’s comfort and abilities. In addition, fatigue and stress are risk factors for the deterioration of cognitive functions. Not without impact are breathing mixtures, as most so-called inert gases (e.g., nitrogen and helium) remain inert for metabolism but still affect the central nervous system, leading to inert gas narcosis which deteriorates cognitive functions and remains incompletely explained as a phenomenon. Nevertheless, using a heliox mixture makes the diver prone to hypothermia, which is aggravated in cold water. Therefore, these factors impair cognitive performance to the greatest disadvantage for the diver and his or her safety [9].

However, the latest publications bring conclusions regarding negative effects on cognitive functions. Most of them are defined as acute and reversible, if we assume rare diving performances as a summer leisure activity. Still, referring to regular dives at greater depths (over 40 m), we cannot rule out the risk of chronic effects on the nervous system and cognitive functions. The effects of diving depend on the depth of immersion together with an increase in pressure, time of exposure, breathing mixtures, temperature of the water and kind of physical activity performed under the water [10].

Below, we summarize the possible side effects of diving on cognitive functions with alignment to acute and chronic effects together with division into recreational and saturation diving, including serious ailments such as high-pressure neurological syndrome (HPNS).

## 2. Acute Effects of Recreational Diving on Cognitive Functions

Many published reports highlight the negative effects of diving, in general, on the central nervous system. Apart from rapid ascent, other factors can influence the safety of a diver underwater. Among such factors, we particularly highlight exercise as a type of physical effort, stress, environmental factors (such as water temperature and visibility) and mixture of inhaled air. Psychological and environmental factors including cold, water currents, visibility, stress and claustrophobia can alter cognitive functions [11]. Environmental factors can diminish the ability to properly assess underwater situations and cause the deterioration of the diver’s skills [12].

Highlighting the meaning of cold in cognitive disorders, we should list cold water as an environmental stress that affects cognitive functions [13].

Although Martin et al. [13] mentioned that researchers [14,15,16] did not find a direct cold water correlation in memory or executive function disturbances, a decrease in correct task executions was found in other studies when comparing cold and control conditions. What is more, a decrease of up to 23% was observed when the water temperature was lowered from 23 °C to 10 °C. The above studies examined the effect of cold on cognitive functions during military task performance. What remains unexplained is that tasks that were well trained before were not greatly affected. However, cold acclimation, in the format of repetitive cold exposure over 10 days, did improve the results of tasks concerning working memory. A cold ambient temperature together with cold water (<25 °C) and prolonged time spent underwater leading to diver cooling may direct the diver off-task. Knowing that heliox, one of the breathing mixtures used, contains helium, characterized by a higher thermal conductivity coefficient, extra heat loss should be considered when estimating the diver’s cooling in association with its influence on underwater cognitive functioning. However, the researchers found that the degree of cognitive disorder depends on the exposure time. More affected cognitive functioning was observed in divers that stayed in the cold water for over 30 min [13].

Water visibility as another environmental factor seems to have an indirect effect [17]. Worse visibility impedes spatial orientation and distance assessment and may increase stress during diving. Together, with all of the factors mentioned above, clear water makes the dive safer and more comfortable. However, diving in lakes and brines cannot provide the diver with perfect diving conditions. In addition, ocean currents together with visibility may affect the judgment of underwater situations as a component of environmental stress [11].

Research describing a correlation between physical activity and cognitive functions was published recently by Möller et al. [18]. A group of 27 volunteer divers was tested before the exercise, after 20 min of underwater activity (in a water temperature of 28 °C, mild physical activity taken at a depth of 3.8 m or high physical activity performed at a depth of 5 m) and afterward in dry ambient surroundings (in the laboratory). The breathing mixture consisted of 21% oxygen and 78% nitrogen as a standard breathing gas for scuba divers. The authors assessed working memory and inhibitory ability. The results indicate that a short (20 min) time of mild physical activity in submersion enhances cognitive performance, but only in working memory. Shoemaker et al. [19] described the effects of physical exercise performed in the water. The breathing mixture consisted of 21% oxygen and 78% nitrogen as a standard breathing gas for scuba divers. Firstly, physical activity increases cerebral blood flow and cognitive performance in the acute phase. However, the type of activity (breast-stroke swimming versus prone position underwater) with muscle involvement is influential. The researchers showed that the intensity of physical exercise can alter the mean cerebral artery blood velocity through changes in exhaled pCO_2_, impacting cognitive functioning. In addition, the duration of water immersion is meaningful as immersion for up to 20 min does not affect cognitive functions [19].

Inert gas narcosis is the most frequently observed acute effect of diving without described long-term results [20]. Nevertheless, symptoms of nitrogen narcosis are observed in hyperbaric conditions but are usually minimized when normobaric conditions are restored [10]. So far, no long-lasting side effects of nitrogen narcosis have been described; however, Balestra et al. reported persisting deterioration of CFFF, even after 30 min post-exposure [21]. Pathogenesis of this phenomenon is based on breathing mixtures’ partial pressure. In normobaric conditions, breathing air contains oxygen 21% and 78% nitrogen. In hyperbaric conditions, consequently with an increase in ambient pressure, the partial pressure of nitrogen also increases. The effects of the above are changes in behavior, mood improvement, and disturbances of consciousness and neuromuscular functioning. That is why nitrogen narcosis is also called “rapture of the deep” or the “Martini effect”, as a frame of mind is comparable to the feeling after one glass of a martini drink; symptoms of this phenomenon are described while diving at the depth of 15 m or after general anesthesia. The exact explanation is still discussed, but the narcotic effects mentioned above are probably correlated with the hypothesis that the solubility of nitrogen and anesthetics in lipids are similar and so are the effects on the central nervous system [21]. The other theory explains this phenomenon as a disruption in inhibitory channel activity through the gamma-aminobutyric acid neurotransmitter and its transmembrane receptors. Therefore, it can be compared to driving a car after having drunk an alcoholic drink; nitrogen narcosis causes blurred vision, uncontrolled behavior, prolonged time of reaction and motor discoordination [10]. It is also reversible in normobaric conditions, as it affects cognitive functions while diving as a risk factor.

Linking cold water immersion, long diving times, a lack of visibility, exposure to inert gas narcosis, increasing heart rate with stress and higher catecholamine concentrations, it is not surprising that a panicked diver will not control his or her reactions, not to mention cognitive functions which are supposed to propel survival instincts. Assuming that stress and panic attacks accompany various activities, they may affect not only diving skills, but also other physical activities in particular individuals. Healthy people and those suffering from mental health disorders could have impaired cognitive functions. In a systematic review, Alves et al. [22] compared particular cognitive functions during panic attacks. Most neuropsychological tests showed working and spatial memory impairment with the deterioration of visual memory tasks. Alves et al. [22] highlight that other cognitive functions such as affective and psychomotor processing, attention and executive functions are not clearly affected in panic attacks.

Stress, as an uncontrolled emotion, particularly for inexperienced divers, may lead to underwater incidents. Emotional stress could be associated with the environmental factors mentioned above. Nevertheless, the diver’s personality with an individual ability to cope with new situations and psychical attributes may play an as-yet-undefined role in the prevention of diving accidents [23,24]. Researchers observed that new and difficult tasks are performed more poorly by anxious divers. The intensity of stress factors leading to physiological changes in homeostasis includes an elevation of blood pressure and heart rate caused by the increase in cortisol and adrenocorticotropic hormones. Serum levels of these stress hormones were observed after exposure to increasing environmental pressure together with long-lasting exposure to emotional stress [25]. Martin et al. [13] highlight the role of diving stress in psychomotor performance in relation to cognitive functioning alterations.

Nevertheless, Beneton et al. [26] drew the conclusion that recreational diving practice may help in dealing with stress. Assuming the necessary preparation for recreational diving, besides physical training in breathing, an ability to maintain mindful functioning seems to be crucial in new, stressful situations underwater. A group of volunteers was investigated with a perceived stress scale, mindfulness inventory and profile of mood states. The results showed that recreational diving, as an activity taken occasionally during summer holidays, decreases perceived stress and improves mindfulness abilities. Lee et al. [27] published a study comparing the effects of diving on the cardiovascular system and cognitive functions, depending on breathing gas. Both attempts assumed diving at a depth of 26 m for over 20 min in one-week intervals. The first dive with compressed air (oxygen 21% and nitrogen 79%) and the other with heliox (oxygen 21% and helium 79%) as breathing mixtures were registered with the assessment of O_2_ blood saturation, lactate level, heart rate, blood pressure and pulse wave. In relation to the cardiovascular system, no significant differences were observed. However, in neuropsychological tests, the results showed that diving with heliox as a breathing mixture has less of an effect on cognitive functions and can even improve them. Comparing the results of the Stroop test after each dive, volunteers reported that attempts with heliox increased cognitive functions such as processing speed and time of reaction. Together with better O_2_ blood saturation while breathing with heliox, the increased oxygenation of brain tissues seems to be safer for the diver to avoid negative effects on cognitive functions, such as their deterioration.

## 3. Carbon Dioxide/Oxygen Poisoning and “Inert Gas” Effects

Carbon dioxide, which is exhaled from the human body with each breath as a product of cellular metabolism, can have poisonous effects in the same way as the gases mentioned above. The impact of CO_2_ on cognition has been extensively investigated in indoor office environments and simulated spaceflights, with excellent reviews covering both topics [28,29,30,31].

Gill et al. investigated working memory performance using the n-back test in normoxic (0.21 ATA O_2_) and hyperoxic (1.3 ATA O_2_) conditions with and without CO_2_ (exposure within individual CO_2_ tolerance limits). CO_2_ exposure caused a deterioration in n-back test results in both normoxia and hyperoxia [32].

Freiberger. et al. [33] assessed the influence of respiratory gases in hyperbaric conditions on psychomotor functions in a group of 42 male, U.S. Navy divers. The study group was neuropsychologically tested at rest and during exercise with a power of 100 watts. The results were compared depending on breathing gas pressures, which included N_2_ 0, 4.5 ATA, 5.6 ATA and O_2_ 0.21 ATA, 1 ATA, 1.22 ATA. In addition, tests performed during exercises were assessed under the influence of CO_2_ 0.075 ATA compared to no CO_2_ in the breathing mixture. The observations proved that high nitrogen partial pressure (4 times higher than atmospheric pressure) weakens working memory, alertness and planning but does not affect psychomotor performance in this group. However, hyperoxia intensifies nitrogen narcosis and memory impairment. In addition, hypercapnia, per se, may slow cognitive performance, but not as much as nitrogen narcosis. Freiberger et al. [26] observed that the impact of nitrogen, carbon dioxide and oxygen may differently impair cognitive functions.

In normobaric conditions, when air is breathed, nitrogen as a major component of ambient gas inhaled is an inert gas in the human body. Among the inert gases, besides nitrogen, are krypton, argon, helium and xenon. Those in the human body do not affect physiology as they do not take part in gas exchange and cellular respiration. However, in hyperbaric conditions, narcosis symptoms are observed due to the higher partial pressure of those, so they are all reported in inert gas narcosis [10]. The effects of anesthetics and inert gas narcosis are compared due to interactions with ion channel neurotransmitter receptors. Observed symptoms include confusion; a longer time of reaction; and disturbed concentration, coordination and memory. In addition, it is important to consider gas toxicity, particularly carbon dioxide and oxygen toxicity, which is exacerbated in high pressure conditions. Altered pressure of nitrogen and oxygen leads to hyperoxia and the formation of nitrogen bubbles from dissolved circulating blood. That manifests through tinnitus, vertigo, headache, inner ear damage, joint pain, pulmonary embolism and spinal cord damage (e.g., lower limb paresis) [10]. Nevertheless, gas may be present in the form of microbubbles with no clinical implications. What should be highlighted is that neurological symptoms such as those mentioned above are a clinical manifestation of accumulated gas bubbles. Meanwhile, microbubbles may occur without the described symptoms of neurologic function neglect [34]. Steinberg and Doppelmayr [9] investigated executive functions among divers after dives at 5 and 20 m. The subjects were breathing with normal air in normobaric conditions with a pressure-adjusted amount of air for one minute (breathing mixture consisted of 21% oxygen and 78% nitrogen as a standard breathing gas for scuba divers). The results showed partial cognitive impairment. After dives at 5 m, no significant differences in cognitive functioning were observed. However, diving at 20 m revealed the inhibition of self-control ability that is indispensable in sudden, underwater situations. The researchers bind these results with nitrogen narcosis effects that impair executive functions at a depth of 20 m.

## 4. Chronic Effects of Recreational Diving on Cognitive Functions

In medical fields, the definition of “chronic” means something that persists for more than 3 months. Published reports raise a suspicion that diving, particularly when regularly performed, may have chronic effects on the nervous system and on cognitive functions. It should be highlighted that the most commonly described effects called “chronic” are observed sooner than 3 months after diving. It is known that long-term effects on cognitive functions, manifesting in spatial disorientation and impaired memory, are more often observed in middle-aged divers that perform regular, frequent dives (more than 100 dives/year) at depths greater than 40 m [35]. The exact mechanism of the above observations remains uncertain, with a lack of clarity around whether it is a subclinical manifestation of decompression disease or not.

Hemelryck et al. [36] described lesions in the central nervous system related to diving. The study group consisted of active divers and boxers. As a group of divers, young volunteers were included at ages of less than 45 years, with at least 200 performed dives and with no history of decompression sickness. The boxer group included professional boxers with no history of serious brain injury that would exclude them from fights for longer than 3 months. The study group was examined using brain magnetic resonance imaging (MRI). By visualizing long-term effects, the researchers found lesion-like signals in the brain on MRI scans; however, no vascular etiology was found in radiological imaging. Most lesions were described in the white matter, so the conclusion links their locus with cognitive disturbances, particularly those related to coordination and associated motor abilities. What is interesting is that lesions described in MRI were compared by radiologists to multiple sclerosis abnormalities without a vascular basis. The researchers combined this with gas emboli that may occur often among divers as low-intensity or symptomless DCS. A study group in this research contained divers with no history of DCS and performed neuropsychological tests checking the short-term memory, coordination, attention and time of reaction. The results were compared to control groups and boxers that are particularly exposed to head injuries and chronic brain damage. Among the recreational divers, long-term effects on short-term memory have been observed. However, exacerbation of short-term memory was not associated with a history of decompression sickness or symptomatic cognitive function impairment. As higher cognitive functions, we include all control processes with organizing abilities and skills of adaptation to new, also environmental, conditions. In summary, higher cognitive functions refer to, inter alia, a survival instinct that helps us to manage problems in new situations. In addition, such an ability is crucial to effectively solve unpredictable situations during underwater activities.

Based on the above definition, Hemelryck et al. assessed the time of reaction, attention, perception, coding and memory among recreational divers, together with the above-mentioned MRI testing. The results revealed that the perception and coding were worsened among divers. However, the same results were observed in the group of boxers. Both of these groups saw worse results in the neuropsychological tests compared to a control group. As a reason for these results, the researchers state that obvious nitrogen and hyperoxia effects, as well as human physiological changes in hyperbaric conditions, should be considered. However, regarding adaptive abilities and the plasticity of neurons, symptomless or low-intensity DCS may be explained in regular recreational diving [37].

Nevertheless, considering nitrogen narcosis, observed during recreational dives, this manifests as cognitive function disturbance, such as prolonged time of reaction, aggravated coordination and memory. It is interesting that, in relation to the above-mentioned neuronal plasticity, it is still observed during each repeated dive as an acute effect of diving on cognitive functioning. Neurotransmitter changes are probably responsible for this effect. However, more studies have to be performed to further investigate this phenomenon.

Slosman et al. [38] carried out a cohort study investigating the influence of regular dives, depending on diving depth and water temperature, on cerebral blood flow, cognitive functions and behavior. The researchers found a correlation between depth and number of dives, environmental factors and individual factors such as age and body mass index in the group of 215 divers. Healthy recreational divers were diagnosed with radiological imaging and neuropsychological tests. The results showed that cold water and deeper dives have negative effects on cerebral blood flow and cognitive functions. In the group of examined volunteers, prolonged times of reaction, weakened decision making and a slowdown in performing tasks were observed. Based on radiological imaging, the observed cognitive disturbances were connected to lesions in the brain white matter. Most chronic neurological diseases (including neurodegenerative maladies) are explained by white matter lesions, which seem to have similar etiology in chronic effects on cognitive functioning.

The conclusions of Slosman et al. are that regular recreational diving has chronic (long-term) effects on the nervous system. By the mechanism of cerebral perfusion changes, neuropsychological functions are affected. In addition, greater deterioration is observed when diving conditions are more extreme (such as in cold water or at diving depths over 40 m). In addition, performing more than 100 dives per year was associated with advanced cognitive function aggravation [38].

## 5. Acute and Chronic Effects of Professional Diving on Cognitive Functions

Meckler et al. investigated whether the response-selection phase of sensorimotor information processing is impacted by inert gas narcosis. The hyperbaric condition, in which 10 participants (French Navy divers) were exposed to 6 absolute atmospheres of 8.33% O_2_ nitrox, was contrasted with an air normobaric state. Subjects in both situations completed a between-hand choice-RT task in which the stimulus–response association rule was specifically manipulated. Inert gas narcosis altered the effect of this manipulation, which is intended to affect response-selection processes. The authors concluded that response-selection processes are one of the factors contributing to the effects of inert gas narcosis on information processing [39].

Ergen et al. compared healthy controls and groups of air divers regardless of their experience [40]. The authors performed an assessment of attention, language, memory, executive functions and visuospatial skills. According to neuropsychological testing, the more experienced group (2001–4000 dives) performed significantly worse on measures of visuo-constructional and visual long-term memory than the healthy (non-diving) and less experienced groups. The less experienced diving group outperformed the healthy group in planning-related tests. The authors concluded that experienced divers showed the majority of the alterations in neurophysiological measurements and a worse neuropsychological performance, which may be read as accumulating long-term effects on brain function related to diving.

## 6. Acute and Chronic Effects of Saturation Diving on Cognitive Functions

Saturation diving is commonly used experimentally and during building underwater constructions. Saturation divers may spend up to a few weeks in the hyperbaric environment, part-time underwater and part-time in a dry atmosphere of either a chamber or caisson. Engineers working on undersea construction sites use this type of diving to stay at specified pressures for a long time. Everyday compression to get to the workplace and decompression afterward would be too risky due to the acute effects of diving such as decompression illness. That is why staying at a specified pressure, so-called “storage pressure”, and then just diving deeper to the workplace for a few hours before going back to the on-surface habitats seems to be safer for the employee.

The number of studies investigating cognitive functions in saturation diving is highly limited. Nevertheless, saturation diving is performed mostly as an occupation. Imbert et al. [41] performed an observational study to monitor vascular changes after decompression. Gas emboli and flow-mediated dilatation seem to be associated with decompression sickness. In this study, vascular changes were observed after saturation. However, all divers (49 examined saturation divers) achieved a full recovery after noted impairments. The authors concluded that vascular changes are associated with inflammatory and oxidative stress due to diving conditions.

Hou et al. [42] investigated mental and cognitive functions in saturation divers during simulated 480-m diving with heliox (helium and oxygen) as a breathing mixture. Four young, healthy and right-handed men were included in this study. Testing was performed before decompression (0 m); during compression, stay and decompression; and 3 days after and 1 month after the simulation. The results showed that the examined reactions and abilities (hand–eye coordination test, grip strength test, spatial memory and mental rotation) were disturbed. In these experiments, the investigators observed that together with greater depth (compression phase), the time of reaction was significantly longer, and a similar influence of an increase in depth was observed in spatial memory tests. Motor abilities were ambiguously changed. Grip strength was weakened after crossing the compression line of 230 m, but hand–eye coordination was significantly impaired at the depth of 400 m; however, during decompression, it was again identifiable at the depth of 300 m. In the 3-month check-up, no impairments in the field of tested abilities were found. However, chronic effect assessment of saturation diving on cognitive functions seems to be unpredictable so far due to the limited number of reports in this field.

Brubakk et al. [43] described the physiology and pathophysiology in saturation diving. Most of the issues raised did not concern cognitive functions, but together with the influence of environmental factors, they were slightly elaborated. In recent decades, hypothermia has been raised as a direct risk factor in commercial saturation diving. The physiological reaction to cold is shivering, muscle tremors and subjective cold sense. Compressed divers may not observe the proper perception of cold during long stays underwater, leading to the malfunction of thermogenesis and aggravating hypothermia. In addition, divers breathing with a heliox mixture in hyperbaric conditions are more prone to heat loss (physical property of helium). Hypothermia is blended with saturation diver accidents at work. Cold ambient temperature and windy weather decreasing perceived ambient temperature together with heliox as a breathing mixture increase heat loss, leading to cognitive function disturbances. Impaired alertness and eye–motor coordination are life-threatening factors in underwater work.

In addition, dehydration has been raised as a factor influencing cognitive functions in the above-mentioned report. Brubakk et al. [43] state that dehydration deteriorates physical and cognitive performance and may also increase the risk of decompression sickness. The specificity of saturation diving as an occupation requires the employee to stay in particular conditions for up to 3–4 h while working. In addition, sweating in a working suit during physical work increases water loss. As reclaiming is less accessible, ongoing thermogenesis in progress dehydration is impossible to fudge. The authors highlight that proper hydration is essential for the safety of saturation divers as dehydration accelerates bubble formation and increases the risk of decompression disease. So far, according to Brubakk et al., it is still not known whether dehydration is an isolated risk factor of cognitive function perturbation or a common effect of all factors (stress, fatigue, etc.) that together may cause cognitive impairment [43].

Rosen et al. [44] conducted a controlled observational study concerning neuronal damage at a cellular level in saturation divers. The study group of 14 submarine volunteers stayed compressed in a hyperbaric chamber for 36 h; afterward, the decompression lasted for 70 h. Biomarkers such as hemoglobin concentration, albumin and tau protein were taken before and during compression and after decompression. Blood samples were taken from a control group of 12 men. The results showed no significant changes in the assessed parameters. Nevertheless, the researchers raised the fact that albumin level changes together with dehydration, but further studies need to be performed to determine whether lower albumin serum concentrations affect cognitive functions by increasing neurological stress and neurological activity. Examined protein levels such as those of tau, glial fibrillary protein and neurofilaments are considered to be biomarkers of brain damage in traumas and neuronal hypoxia. To date, saturation diving appears to be a safe occupation in long-term (up to 70 h after decompression) observation.

Therefore, based on the current knowledge, it seems that saturation diving does not have a direct influence on long-term cognitive functions. Nonetheless, together with risk factors present in underwater working conditions, cognitive functions can be vicariously affected.

## 7. HPNS—High-Pressure Neurological Syndrome

High-pressure neurological syndrome is often observed during dives at depths of over 100 m [10]. This is mostly described as dysfunctions in neurologic and cognitive states with changes in electroencephalography (EEG). However, the most common symptoms are headache and nausea, with behavioral deviations such as psychoses or dysphoria. Together with tremors, myoclonus, hyperreflexia and typical abnormalities observed in other neurological diseases, it is called neurological syndrome due to the effect of high-pressure influences on the nervous system. The onset of HPNS is earlier if the compression rate is fast, and full-blown disease is observed if the dive is prolonged, especially for deep dives. Nevertheless, symptoms of HPNS are reduced when decompression is conducted.

As the nervous system is the most sensitive to any changes in the human body, the most severe disturbances due to increased ambient pressure will present in neurological syndromes. Of interest, neurological syndrome is usually observed during diving at of depths 100–120 msw (0.1 MPa) and deeper. Knowing that standard diving courses consist of diving up to 40 msw, we would not expect HPNS among the diving group participants. HPNS has been anagenetically reported in extreme recreational diving [45,46], but no scientifically approved observations have been made in the group of recreational divers to date. However, what should be highlighted here is that with current technology improvements, dives to 100 m and beyond are becoming normalized for hobby divers. Still, no relevant data have been collected about this new diving trend.

The pathophysiology of HPNS includes changes in ionic channels and pumps due to high pressure affecting ionic constants in cellular membranes [47]. Changed conduction of these transporters reduces their amplitude and their conducting velocity. Transferring that to tissues, increased excitability together with reduced conductivity results in a delay in neuronal information transmission. Therefore, all possible neurological symptoms during the dive can be observed in high-pressure neurological syndrome [48].

Vaernes et al. [49] described high-pressure neurological syndrome in a group of 18 divers. However, this trial included hyperbaric compression with heliox together with neuropsychological and neurophysiological assessment. Reported pressures were 240 msw (2.4 MPa) and 360 msw (3.6 MPa). The results showed that most of the volunteers had moderate symptoms of HPNS. In one-third of assessed divers, EEG changes were observed, with the most commonly noted being tremors and cognitive dysfunctions, such as impaired memory and motor response time, with no significant motor disability.

Talpalar [47], in a systematic review, described previous general knowledge about HPNS. Despite the presentation of motor and sensory disabilities, mood changes and sleep disorders have been observed in what seems to be a kind of vicious circle. Sleep deprivation as a potential risk of decompression sickness becomes a symptom of mild HPNS. In severe courses, sleepiness is observed. Similar to other neurological manifestations, the images of mild and severe courses may differ, aggravate or be opposite. However, what has to be mentioned is that motor, sensory and autonomic dysfunctions usually worsen together with severity. Sleeping disorders show the opposite clinical manifestations together with illness advancement.

In addition, Talpalar [47] points to the significance of breathing mixtures used by divers. Types and proportions of breathing gases define the severity of HPNS. Bennett and Vearnes [49,50,51] describe trials with heliox (breathing mixture containing helium and oxygen) and trimix (mixture containing helium, oxygen and nitrogen or hydrogen/or argon). As nitrogen at high pressure is well known for its toxic effects, all attempts were focused on reducing its proportions in mixtures.

In recent years, more trials and observations have been conducted, mostly concentrating on diving techniques to decrease the risk of developing HPNS in deep dives. Balestra et al. [52] refer to the adaptive potential of the brain and human body in response to extreme environments. In particular, neuroplasticity and cognitive functions need to be investigated under extreme environmental conditions.

## 8. Conclusions

Recreational diving may seem to be a safe sport from the viewpoint of tourism. The number of seasonal dives is increasing each year, and so is the number of diving centers offering single dives for summertime tourists. Knowing that its acute effects on cognition are reversible, no chronic changes in cognitive functioning should be observed as long as this activity is taken seriously. Chronic effects of regular, long-term dives at depths going beyond recreational diving may include some changes in cognition. However, more studies with well-designed neuropsychological and neuroimaging protocols would be helpful to gather more data and enable further conclusions to be drawn in this field. In particular, studies concerning decision making under hyperbaric conditions should be performed, as this is crucial in life-threatening situations underwater. The question about decision making during diving activity remains unanswered. Undoubtedly, it may be key in the prevention of diving accidents. The effects of diving types and breathing mixtures on cognitive functions are summarized in Table 1 and Table 2.

## Figures and Tables

**Table 1 biology-12-00229-t001:** The influence of diving types on cognitive functions.

Cognitive Function	Acute Impact of Recreational Diving	Chronic Impact of Recreational Diving	Acute Effect of Saturation Diving	Chronic Effects of Saturation Diving
Alertness	+/−	+	+/−	no data
Sensing	no data	no data	+/−	no data
Time of reaction	+/−	no data	+/−	no data
Perception	+/−	+	+/−	no data
Memory	+/−	+	+/−	no data
Learning	no data	+	no data	no data
Thinking	+/−	no data	no data	no data
Decision making *	no data	no data	no data	no data
+/(−) transient effect; (+) observed cognitive function disturbance				

* Decision making is an indispensable cognitive function for propelling survival instinct in life-threatening situations that may be unavoidable in any underwater activity.

**Table 2 biology-12-00229-t002:** Summary of studies performed on various breathing mixtures and their effects on cognitive functioning.

Research	Breathing Mixture	Examined Initiator	Effects/Conclusions
Lee et al., 2020 [27]	compressed air (21% oxygen + 78% nitrogen) versus heliox (21% oxygen + 79% helium)	cognitive functioning	heliox affects CFs less and can even improve processing speed and reaction time
Shoemaker et al., 2019 [19]	standard breathing gas for scuba diving (21% oxygen + 78% nitrogen)	general effects of physical exercise	the improvement of CFs in acute phase, no significant changes in 20 min immersions
Moller et al., 2019 [18]	standard breathing gas for scuba diving (21% oxygen + 78% nitrogen)	type of physical activity—working memory and inhibitory activity	short-time mild physical activity enhances working memory
Steinberg and Doppelmayr, 2017 [9]	standard breathing gas for scuba diving (21% oxygen + 78% nitrogen)	cognitive functioning	inhibition of self-control ability while diving at 20 m depth
Freiberger et al., 2016 [33]	nitrogen pressures and oxygen pressures in breathing mixtures	psychomotor functions	high nitrogen partial pressures impair working memory, alertness and planning

CFs—cognitive functions.

## Data Availability

Not applicable.

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
