# Peer review of "Cognitive Functions in Scuba, Technical and Saturation Diving"

_biology, 2023, doi:10.3390/biology12020229_

Round 1

Reviewer 1 Report

Line 40, Nitrous Oxide is not correct, Nitrogen more appropriate definition - Line 79, saturated diving not a usual definition, saturation diving preferable -Line 241, a reference would be better to support the statement -                 Line 271, "air" bubbles should be better defined as "gas" bubbles -                Line 489, compression rate is "frequent", I believe the authors intended "fast"                                                                                                                           Line534, I believe authors intended "extreme" environment conditions, not "supreme"

General comments: the paper is interesting as a comprehensive review of both acute and chronic adverse effects of diving in its several modalities, from recreational to professional. My impression is that the paper focuses on relatively well known physiological effects but less than expected, by reading the title, on psychological and cognitive effects. I believe it would be better focusing on the latter, expanding the related sections and references while reducing the parts more oriented to describing known physiological and pathophysiological effects. 

I also suggest extensive English language revision since both construction, occasionally, and more often terminology would benefit from it and the overall value of the manuscript would improve

Author Response

REVIEW REPORT 1

Thank you for your report. I appreciate your time and suggestions. Proposed changes has been made and all inappropriate words have been replaced as advised.

  1. Line 40, Nitrous Oxide is not correct, Nitrogen more appropriate definition.

Thank you, wrong name has been changed to correct term.

  1. Line 79, saturated diving not a usual definition, saturation diving preferable.

I have replaced it with proper name, thank you.

  1. Line 241, a reference would be better to support the statement.

Thank you. We have added references of Van Wijk et al.

(Van Wijk CH, Martin JH, Meintjes WA. Diving under the influence: issues in researching personality and inert gas narcosis. Int Marit Health. 2017;68(1):52-59. doi: 10.5603/IMH.2017.0009. PMID: 28357837.

Van Wijk CH. Personality and behavioural outcomes in diving: current status and recommendations for future research. Diving Hyperb Med. 2017 Dec;47(4):248-252. doi: 10.28920/dhm47.4.248-252. PMID: 29241235; PMCID: PMC6706337)

  1. Line 271, "air" bubbles should be better defined as "gas" bubbles.

The term “air” has been replaced as advised. Thank you for this suggestion.

  1. Line 489, compression rate is "frequent", I believe the authors intended "fast"      

Incorrect adjective has been replaced. Thank you.   

  1. Line534, I believe authors intended "extreme" environment conditions, not "supreme"

That is right. Inappropriate word was used, we changed it.

  1. I also suggest extensive English language revision since both construction, occasionally, and more often terminology would benefit from it and the overall value of the manuscript would improve.

The manuscript has been professionally proofread (Proof-Reading-Service.com)

Kind regards,

Rita Sharma

Reviewer 2 Report

Thank you for the opportunity to review this promising manuscript. Unfortunately, there are many factual errors within it, such that it is difficult to know where to start. If the authors can cite some evidence to support their assertion that diving is increasing in popularity then all well and good, otherwise there is no basis for the first sentence of the manuscript. Instead I think the unsupported point can be made that "Diving is a popular recreational activity", though what is meant by "popular" remains contentious. I would not cite an unreliable DEMA webpage that has not been peer-reviewed, when there are plenty of peer-reviewed papers that show diving is undertaken by millions of people, for example Buzzacott, P., et al. (2021). "Mortality rate during professionally guided scuba diving experiences for uncertified divers, 1992-2019." Diving Hyperb Med. 51(2): 147-151 or Buzzacott, P., et al. (2022). "Health and wellbeing of recently active U.S. scuba divers." Diving and Hyperb Med 52(1) to name but two that come to mind. Line 34, the number of travellers has NOT tripled in the last 40 years. If it has then cite some evidence. First reports describing DCS were made after the construction of the Eads and Brooklyn Bridges, and then the Hudson River Tunnel in 1889. I question the necessity of using Roman numerals in lines 36/37. Decompression sickness is NOT caused by bubbles of nitrous oxide (line 40). The above comments address just the first 12 lines of the manuscript. I strongly urge the authors to seek some help when revising this manuscript, which is unpublishable in its current form. Indeed, I would be surprised if Jacek Kot gave his approval to submit it as is.

Some other more general comments I have are that, overall, the manuscript would benefit from an edit by a native speaker of English. SCUBA no longer needs to be spelled out in full as it is now recognised as a word in its own right. Contrary to line 81, there have been numerous other reviews of cognitive dysfunction caused by breathing inert gas under pressure, including  

1.           Bennett P. The physiology of nitrogen narcosis and the high pressure nervous syndrome. In: Strauss RH, ed. Diving Medicine. Grune and Stratton; 1976:157-180:chap 12.

2.           Bennett P. Inert gas narcosis and high-pressure nervous syndrome. In: Bove AA, ed. Bove and Davis Diving Medicine. Saunders; 2004:225-239.

3.           Brauer JR. The effects of a hyperbaric environment on information processing in recreational divers as measured by a table-reading task: A pilot study. Dissertation Abstracts International: Section B: the Sciences & Engineering. 1996;56(11-B),

4.           Clark JE. Moving in extreme environments: inert gas narcosis and underwater activities. Extrem Physiol Med. 2015;4:1. doi:10.1186/s13728-014-0020-7

5.           Fowler B, Ackles KN, Porlier G. Effects of inert gas narcosis on behavior--a critical review. Undersea Biomed Res. 1985;12(4):369-402. http://www.ncbi.nlm.nih.gov/entrez/query.fcgi?cmd=Retrieve&db=PubMed&dopt=Citation&list_uids=4082343

6.           Grover CA, Grover DH. Albert Behnke: nitrogen narcosis. J Emerg Med. 2014;46(2):225-227. doi:10.1016/j.jemermed.2013.08.080

7.           Lowry C. Inert gas narcosis. In: Edmonds C, Lowry C, Pennefather J, Walker R, eds. Diving and Subaquatic Medicine. 4th ed. Edward Arnold; 2002:183-193:chap 15. 

Line 356 - Slosman included all the recreational diving instructors in his sample in the cold-water group, which introduced a systematic bias into that analysis because in those days recreational diving instructors were required to teach controlled emergency swimming ascents at a ratio of 1-to-1, meaning an instructor would make repeated emergency ascents during training. Up, down, up, down, over and over, one student at a time.

Line 482 - I have been very active in recreational/technical diving for 30 years and I've never heard HPNS referred to as "blue hole".

Lastly, the abstract does not follow the layout and flow of the manuscript. It highlights technical diving as an activity worth examining, twice, but then the main text makes only a brief mention instead of a dedicated section on technical diving. Dives in excess of 100m depth are made daily in various places in the world, they are all but routine nowadays, dives to 150m+ are regularly made for example by cave divers in France, and in New Zealand dives in 7C water are made each year in excess of 200m depth. My understanding is that this year Hydrogen is being introduced into the breathing mix.

So, in summary, I encourage the corresponding author to seek out assistance and to heavily revise this manuscript. I will be very happy to review it again if required, I feel it has the potential to make an original contribution to what is known on the topic. Please also see https://www.mdpi.com/1648-9144/58/6/739 

Author Response

REVIEW REPORT 2

Thank you for your report. I appreciate your time and suggestions. Proposed changes has been made and all inappropriate words have been replaced as advised. Below, I did my best to respond your comments.

  1. If the authors can cite some evidence to support their assertion that diving is increasing in popularity then all well and good, otherwise there is no basis for the first sentence of the manuscript. Instead I think the unsupported point can be made that "Diving is a popular recreational activity", though what is meant by "popular" remains contentious. I would not cite an unreliable DEMA webpage that has not been peer-reviewed, when there are plenty of peer-reviewed papers that show diving is undertaken by millions of people, for example Buzzacott, P., et al. (2021).

Concerning popularity of diving, I must admit that I based on DEMA data from previous years. However, I understand that this source of data was a bad choice.As a young researcher with not much experience, I particularly value your impact in my manuscript revision. I was aware that my first article may have errors. Thank you for pointing exact publications, after reading them I have extended my knowledge. Also I receive it as a new experience that is necessary in my professional development.

  1. Line 34, the number of travellers has NOT tripled in the last 40 years. If it has then cite some evidence.

If it is a about number of travelers, I focused on internet data before the pandemic as then I started writing my review. I should have updated this data. Not to mention, that now I understand that such sources of data are not reliable. Therefore, I decided to change the introduction.

  1. First reports describing DCS were made after the construction of the Eads and Brooklyn Bridges, and then the Hudson River Tunnel in 1889.

The history of decompression disease was also neglected in my review. I have read more about Moir’s airlock and I realize my mistakes. Thank you for this comment.

  1. Decompression sickness is NOT caused by bubbles of nitrous oxide (line 40).

Thank you, the change has been made.

  1. Some other more general comments I have are that, overall, the manuscript would benefit from an edit by a native speaker of English.

The manuscript has been professionally proofread (Proof-Reading-Service.com)

  1. SCUBA no longer needs to be spelled out in full as it is now recognised as a word in its own right.

Thank you, we must admit that spelling it out seems to be sensless.

  1. Contrary to line 81, there have been numerous other reviews of cognitive dysfunction caused by breathing inert gas under pressure (…)

The last sentence from the introduction about being the first review in this field was not meant to deny work of other researchers. Nitrogen narcosis and inert gas narcosis has been widely described already. However, thank you for making us clarify the purpose of this sentence. To make it clear for readers, we decided to delete it.

  1. Line 356 - Slosman included all the recreational diving instructors in his sample in the cold-water group, which introduced a systematic bias into that analysis because in those days recreational diving instructors were required to teach controlled emergency swimming ascents at a ratio of 1-to-1, meaning an instructor would make repeated emergency ascents during training. Up, down, up, down, over and over, one student at a time.

Thank you, I have considered this in the review.

  1. Line 482 - I have been very active in recreational/technical diving for 30 years and I've never heard HPNS referred to as "blue hole".

Thank you, I have read more about HPNS and I admit it was my mistake. Wrong name about this syndrome has been changed.

  1. Lastly, the abstract does not follow the layout and flow of the manuscript. It highlights technical diving as an activity worth examining, twice, but then the main text makes only a brief mention instead of a dedicated section on technical diving.

Our aim was to focus on three particular types of diving (scuba, saturated and technical) with its cognitive functions impairment. Therefore, we changed the manuscript title to make that clear. Thank you.

Kind regards,

Rita Sharma

Reviewer 3 Report

Paper „ Diving and  cognitive functions“ is nice review of acute and chronic consequences of diving on cognitive functions as well as possible risk factors that could provoke it. This topic is of great interest to diving communities that have significant numbers of people and number is increasing (6 milion scuba divers and 40 milion snorklers). It can be also of value to general knowledge on how human body behives under pressure. As far as I am concerned this is the first review in this field and it can be base to future research and of great inportance to divers.

Citations are properly chosen and refer to Authors  claims .

The paper is clear and comprehensive and worth publishing although with some changes. I have some considerations that the Authors will find here below. The numbers of pages and lines refer to the pdf proof.

Major revision:

1). In introduction authors mentioned 40 million snorklers (breath hold divers) as argument of importance of this paper. Breath hold divers are special group of divers which stay underwater for some time in single breath. The breatholding is more and more popular in spearfishing, freedivng competition, shellpicking etc. and usually require out-of-comfort zone activities. Physiology, underwater environment are similar but different enough and deserve explanation in group apart from recreational SCUBA divers, technical divers or saturation divers. Have Authors ment to cover all underwater activities? In that case please include some coments about consequence of breath hold diving on cognitive function. Breath hold divers can    also get cold, dive in low visibilty, be prone to environmental influences, get decompression sickness, nitrogen narcosis and experience great changes in arterial gases from hyperoxia to hypoxemia and from hypocapnia to hypercapnia in single breath therefore can have cognitive functions impairment. Or maybe Authors reviewed only scuba divers, saturational and technical divers? In that case they should change headline .

2.) It is not clear to me if Authors concluded  that diving seems to be safe (line 536) because acute effect on cognitions are reversible and no chronic changes are observed or they think diving is safe in general? Authors mentioned in introduction that „safety of this activities has been discussed worldwide“ (line 35). Serious  diving accidents  are rare but still happen and somethime with deadly outcome therefore the Authors should reconsider the focus of their conclusion.

The greatest value of this review, apart from concluding that recreational scuba diving has no impact on chronic cognitive impairement is how this activitie impacts acute cognitive impairement.

If the cognition refers to problem solving, computation and decision making (line 53) than impaired cognition can lead to inproper problem solving and bad decision making therefore to diving accident. Future research in risk factors and how they affect cognitive function and its managment can play important role in diving accident prevention. This should be stressed in conclusion as well as need for further research because this field is neglected by scientists although it is of great importance.

Minor revisions :

1.     Nitrogen not nitrous oxyde cause decompression sickness ( line 40)

2.      Decompresion sickness (without disease) line 42

Author Response

REVIEW REPORT 3

Thank you for your report. I appreciate your time and suggestions. Proposed changes has been made and all inappropriate words have been replaced as advised. I did my best to respond your comments.

  1. In introduction authors mentioned 40 million snorklers (breath hold divers) as argument of importance of this paper. Breath hold divers are special group of divers which stay underwater for some time in single breath. The breatholding is more and more popular in spearfishing, freedivng competition, shellpicking etc. and usually require out-of-comfort zone activities. Physiology, underwater environment are similar but different enough and deserve explanation in group apart from recreational SCUBA divers, technical divers or saturation divers. Have Authors ment to cover all underwater activities? In that case please include some coments about consequence of breath hold diving on cognitive function. Breath hold divers can also get cold, dive in low visibilty, be prone to environmental influences, get decompression sickness, nitrogen narcosis and experience great changes in arterial gases from hyperoxia to hypoxemia and from hypocapnia to hypercapnia in single breath therefore can have cognitive functions impairment. Or maybe Authors reviewed only scuba divers, saturational and technical divers? In that case they should change headline.

We admit that we had reviewed SCUBA, saturation and technical divers. Therefore we decided to change the title of our manuscript to make that clear. Thank you for this suggestion.

  1. It is not clear to me if Authors concluded  that diving seems to be safe (line 536) because acute effect on cognitions are reversible and no chronic changes are observed or they think diving is safe in general? Authors mentioned in introduction that „safety of this activities has been discussed worldwide“ (line 35). Serious  diving accidents  are rare but still happen and somethime with deadly outcome therefore the Authors should reconsider the focus of their conclusion.

We did not mean do conclude that diving is a safe activity. Thank you for pointing this. We changed the sentences in the introduction and in conclusions to make that clarify this. Also, we highlighted the importance of further research in this this subject. What is more we added references concerning diving accidents.

  1. If the cognition refers to problem solving, computation and decision making (line 53) than impaired cognition can lead to inproper problem solving and bad decision making therefore to diving accident. Future research in risk factors and how they affect cognitive function and its managment can play important role in diving accident prevention. This should be stressed in conclusion as well as need for further research because this field is neglected by scientists although it is of great importance.

Thank you,  we changed the introduction to distingiuch the importance of further reaserach ih this field.

  1. Nitrogen not nitrous oxyde cause decompression sickness ( line 40)

Thank you, that is right. I have replaced it with correct term.

  1. Decompresion sickness (without disease) line 42

This change has been made, thank you.

Kind regards,

Rita Sharma

Round 2

Reviewer 1 Report

Authors answered to my comments and revised accordingly

Reviewer 3 Report

Congratulation to Authors for excellent paper! I have no further revisions